# Peptide-Based Drug-Delivery Systems in Biotechnological Applications: Recent Advances and Perspectives

**DOI:** 10.3390/molecules24020351

**Published:** 2019-01-19

**Authors:** Diego Tesauro, Antonella Accardo, Carlo Diaferia, Vittoria Milano, Jean Guillon, Luisa Ronga, Filomena Rossi

**Affiliations:** 1Department of Pharmacy and CIRPeB, Università Federico II, 80134 Naples, Italy; diego.tesauro@unina.it (D.T.); antonella.accardo@unina.it (A.A.); carlo.diaferia@unina.it (C.D.); vittoria.milano@unina.it (V.M.); 2ARNA, INSERM U1212/UMR CNRS 5320, UFR des Sciences Pharmaceutiques, Université de Bordeaux, F-33000 Bordeaux, France; jean.guillon@u-bordeaux.fr; 3Institute of Analytical Sciences, IPREM, UMR 5254, CNRS-University of Pau, 64000 Pau, France; luisa.ronga@univ-pau.fr

**Keywords:** peptide, peptide backbone structures, drug delivery, peptide self-assembling carriers, active targeting receptors, diphenylalanine, binding peptides

## Abstract

Peptides of natural and synthetic sources are compounds operating in a wide range of biological interactions. They play a key role in biotechnological applications as both therapeutic and diagnostic tools. They are easily synthesized thanks to solid-phase peptide devices where the amino acid sequence can be exactly selected at molecular levels, by tuning the basic units. Recently, peptides achieved resounding success in drug delivery and in nanomedicine smart applications. These applications are the most significant challenge of recent decades: they can selectively deliver drugs to only pathological tissues whilst saving the other districts of the body. This specific feature allows a reduction in the drug side effects and increases the drug efficacy. In this context, peptide-based aggregates present many advantages, including biocompatibility, high drug loading capacities, chemical diversity, specific targeting, and stimuli responsive drug delivery. A dual behavior is observed: on the one hand they can fulfill a structural and bioactive role. In this review, we focus on the design and the characterization of drug delivery systems using peptide-based carriers; moreover, we will also highlight the peptide ability to self-assemble and to actively address nanosystems toward specific targets.

## 1. Introduction

Peptides of natural and synthetic origin are compounds involved in a wide variety of biological roles. They act as hormones, enzyme substrates and inhibitors, antibiotics, biological regulators, and so on. Therefore, peptides play an essential role in biotechnological applications as therapeutic and diagnostic agents. Their advantages depend on the strategy applied to produce them and include biocompatibility, low cost, tunable bioactivity, chemical variety, and specific targeting. Moreover, they are easily synthesized, for example, by using solid-phase peptide methodologies where the amino acid sequence can be exactly selected at the molecular level by tuning the basic units [1]. Although the drawbacks related to their use are referred to as metabolic instability via protease degradation, an improved metabolic stability can be pursued through several chemical approaches aimed to modify the original peptide sequences. Some examples are the introduction of specific coded or un-coded amino acids, d-counterparts, cyclization, and DNA recombinant technology. Recently, peptides achieved resounding success in drug delivery and in nanomedicine smart applications, thanks to these innovative approaches. These applications are among the most significant challenges of recent decades in transporting drugs only to pathological tissues whilst other districts in the body are preserved from side effects. This specific feature allows the reduction of unwanted drug effects and increases the drug efficacy [2].

In peptide-containing aggregates, peptide sequence can fulfill a structural or a bioactive role. In detail, peptides play a structural role when the primary amino acid sequence drives or affects the molecular self-assembly by adding remarkable weak non-covalent bonds, electrostatic interactions, hydrogen bonds, hydrophobic and Van der Waals interactions, and п-п stacking between the side chains. Furthermore, peptides play a bioactive role when the full sequence, or a part of it, is deputed to recognize specific receptors, such as those overexpressed by pathological cells. In this review, we will focus on the peptide ability to self-assemble and on potential applications of peptide based nanosystems for nanomedicine. In addition, we report recent examples of peptides employed as delivery systems of anticancer drugs and/or contrast agents for the imaging of tumor pathologies. Finally, we will describe peptide nanosystems able to actively address the active pharmaceutical ingredients (APIs) toward specific biological targets.

## 2. Peptide Self-Assembled Nanostructures

Peptides are able to gather into assorted nanostructures, including nanotubes, nanofibers, nanospheres, and nanovesicles, supported by their device and self-assembly conditions [3]. Different types and structures of peptides, including cyclic and linear peptides, amphiphilic peptides, and *α*-helical and β-sheet peptides, can self-assemble into nanostructures (see Figure 1).

### 2.1. α-Helical and β-Sheet Peptides

The primary feature in the design and synthesis of peptide based biomolecules regards the peptide backbone arrangement in α-helical and β-sheet secondary structures. This is a consequence of the hydrogen bonding pattern interactions through the amide and carbonyls groups in the peptide backbone. After that, the β-strands turn into a β-sheet self-assembled structure that could be rearranged in parallel or antiparallel arrays, according to the direction of the peptide sequences. The peptide is typically designed to contain repeating amino acid residues and distinct hydrophobic and hydrophilic regions. Consequently, the hydrophobic moiety could be hidden within the self-assembled nanostructure while the hydrophilic area could be better exposed to the solvent (water) environment [4]. Unlike β-sheets, α-helices are formed by single peptide chains, where backbone amide components are intramolecularly hydrogen bonded. This arrangement leads to the exposition of side chains of amino acids on the surface of each helix. Thus, their positioning further facilitates the accessibility of the peptide in the solvent.

The regular α-helical peptides with 2,5 helices are shown to aggregate around each other and their structure evolves in nanofibers [5,6]. These α-helical peptides can also self-assemble into nanofibers if they have at least 30 amino acid residues, through helical coiled-coil structures [7]. The hydrophobic residues could promote the helix oligomerization through hydrophobic collapse.

The β-sheet secondary structures are the naturally-occurring motifs most similar to those which carry on into pay peptide self-assembly [8,9]. The β-sheet determines regular alternating hydrophilic and hydrophobic regions in the peptide sequence. The same structure provides the amphiphilic property to the peptide that drives the self-assembly of β-sheets. For instance, β-sheet peptides (namely the QQR holding sequences) can self-assemble into a pH-responsive hydrogel by means of the side chain’s ion affinity for acidic residues of Glu and Arg. These peptides are soluble in neutral pH conditions and they switch into a hydrogel material, in acidic pH surroundings. This behavior can be rationalized in terms of the rearrangement of antiparallel β-sheet tapes. Those β-sheet tapes are obtained at lower pH values and, afterward, stacked together to form nanofibrils in hydrogels [10]. The intramolecular folding β-hairpin peptides are well represented in the self-assembled sequences in various nanostructures, both in water and in space boundaries. Indeed, the self-assembly of β-hairpins in proteins is carried on by the arrangement of two β-sheets in antiparallel plans. The modulation of pH values reproduces the status in which these materials could be gained and tailored. It is well-described that the fundamental mechanism in hydrogels engendered by self-assembly of the β-sheet hairpin structure strongly depends on the increase of the pH values [11].

The peptide self-assembly processes are also kept up by non-covalent interactions that sometime show the key role in the overall configuration. The non-covalent interactions should be taken into high consideration for this grounds, especially when designing peptide self-assembled nanostructures for drug delivery. Indeed, non-covalent interactions shall be rationally applied in the strategies. These non-covalent interactions are effortlessly unfair by external environments, for instance, pH values, temperature array, and the solvent polarity. Indeed, pH values are critical in peptide sequences richer in charged amino acids, such as Glu, Asp, Lys, His, and Arg, as stated above. As a consequence of the rank, these peptides can exhibit negative or positive shell charges. Then, those peptides can self-assemble into different nanostructures, according to the pH values.

### 2.2. Linear Peptides

Recently, data in the literature have reported that short (below six residues) and ultra-short (referred to dipeptides and tripeptides) linear peptides have the ability to self-assemble into many different nanostructures. This aspect, particular interesting, allows to minimize the synthesis and purification steps and to reduce the cost of the production process [12].

One of the most studied prototypes of self-assembling linear peptides is the ultra-short homodipeptide Phe-Phe (FF) (Figure 2) identified by Gazit and co-workers in 2003 as the smallest region of the Aβ-amyloid peptides (Aβ_1-40_ and Aβ_1-42_) prone to the aggregation [13]. The characterization of FF assemblies via single-crystal X-ray diffraction studies showed that this dipeptide is able to generate ring-like networks with a hexagonal symmetry, promoted by head-to-tail interactions established by its charged *N*- and *C*-terminus. This association trend is further stabilized by “T-shaped” contacts between the phenyl aromatic side chains [14]. Molecular dynamic simulations (MD) corroborated these observations, suggesting the ability of this system to form open ring-like peptide networks in aqueous solution [15]. Additional studies highlighted the structural versatility of this motif by showing that, despite its molecular simplicity, FF homodipeptide is able to form more complex supramolecular architectures [16]. Interestingly, simple modifications of the charged state at the *C*- or *N*-terminus of FF, strongly affected its self-assembling pathway. Indeed, the introduction of a thiol group or of a Fmoc-fluorenylmethyloxycarbonyl group can alter the self-assembling phenomena, for example [16].

Nanotubes, nanowires, nanofibrils, spherical vesicles, and organogels are just a few examples of the new peptide materials, based on FF self-assembly. These materials exhibit mechanical properties [17], electrical properties [18], electrochemical properties [19], or optical properties (photoluminescence [20,21] and optical waveguide [22]) properties. In this contest, it is significant to highlight the main property of FF self-assembled nanotubes referring their thermal stability, which is the distinctive skill in bioinspired materials [23]. All these physicochemical characteristics make them suitable for several applications in nanomedicine (tissue engineering, drug delivery, and bioimaging) [18,24,25] and in nanofabrication fields (biosensors, nanodevices, and conducting nanomaterials) (Figure 2) [26,27].

In 2013, Alves et al. suggested for the first time FF-based microtubes (FF-MTs) as potential drug delivery vehicles. In their studies, the authors used Rhodamine B (RhB), a common dye, as a model drug. Data suggested the low in vitro toxicity of the FF-MTs and the potential of these carriers to deliver drugs at constant rates [28]. Beyond the low in vitro toxicity, FF-MTs units showed very high thermal stability (up to 120 °C) and stability towards to the protease degradation. Self-assembled nanotubes obtained for aggregation of unnatural fluorinated-peptides containing two aryl units were found able to penetrate the cultured primary human smooth muscle cells and to locate in their cytoplasmic/perinuclear region [29]. Successively, FF-micro and nanotubes, covalently conjugated to folic acid/magnetic nanoparticles (FA/MNPs), were also evaluated as a potential delivery systems of the anti-cancer therapeutic 5-fluorouracil (5-FU), and of the anti-inflammatory cargo flufenamic acid (FFA) [30].

Analogously to micro- and nanotubes, FF peptide-based nanofibers have also been recently proposed as vehicles of hydrophobic drugs, like hydroxycamphothecin (HCPT), for cancer therapy. In this study, the replacement of some l-amino acids with d-counterparts permitted to further improve the in vitro and in vivo biostability of these peptides against the hydrolysis catalyzed by endogenous peptidases. The protease stability allows for prolonged therapeutic effect with reduction of a tumor mass in a rat model [31]. At the same time, nanofibers of d-peptides generated via enzymatic dephosphorylation were also investigated for the controlled release of the anticancer drug taxol, and of a fluorophore (e.g., 4-nitro-2,1,3-benzoxadiazole) used as imaging agents in vivo [32]. Moreover, the combination of FF with others organic/inorganic molecules brought to the formation of novel hybrid smart materials responsive to the external stimuli, such as pH, enzyme, and oxidative stress. For example, aldehyde molecules can induce cationic diphenylalanine to assemble into biocompatible and biodegradable enzyme-responsive nanocarriers. These nanocarriers loaded with doxorubicin have been proposed as intelligent antitumor agents [33]. Another example of hybrid stimuli-responsive biomaterials is represented by magnetic hydrogel generated for co-assembly under mild conditions of FF with polydopamine spheres coated with Fe_3_O_4_ magnetic nanoparticles [34]. Successively, in 2016 Alves et al. reported the formulation of hybrid materials obtained by conjugation of electrospun polycaprolactone (PCL) fibers and micro/nanotubes of l,l-diphenylylalanine (FF-MNTs). This biodegradable matrix allows the achievement of a stable release of lipophilic anesthetic benzocaine over periods of up to ≈13 hours, much higher than commercially available scaffolds [35].

Concurrently, Wu et al. reported FF-based hybrid nanospheres responsive to pH- and GSH-stimuli. In these spheres, natural alginate dialdehyde (ADA) was employed as cross-linker to induce self-assembly of FF and in situ reducer of Au^3+^ ions into Au nanoparticles (Au NPs). These biocompatible spheres were proposed as drug loading and delivery systems. Indeed, they were found able to encapsulate more than 95% of hydrophobic chemotherapeutic drug (camptothecin, CPT). CPT-loaded spheres exhibited satisfactory stability under normal physiological conditions and excellent pH- and GSH-responsive release at pH 5.0 with 10 mM GSH, which is similar to the tumor microenvironment. Moreover, these nanocarriers can be taken up by cancer cells and have greater cytotoxicity than free drugs [36].

In 2006, Ulijn and co-workers identified Fmoc-FF as one of the first dipeptides able to form a homogeneous, transparent, self-supporting hydrogel with fibrous nanostructure under physiological conditions [37]. The supramolecular nature of Fmoc-FF aggregates suggested its potential use in biological applications, such as controlled drug release, tissue engineering and cell culture. Many examples of multicomponent hydrogels as biocompatible drug delivery systems have been reported until now. FF based hydrogels have been also proposed as vehicle for the delivery of two complementary anticancer drugs, dexamethasone, and either taxol or dehydro-CPT. These peptide hydrogels showed a high in vitro biocompatibility for concentrations up to 100 µM over 48 hours. Moreover, their principal advantages are the improved stability of the drugs over time at 37 °C (i.e., 48 h for the taxol-derivative, and over two weeks for the others) and the slow drug release [38]. Beyond the encapsulation of anticancer drugs, peptide hydrogels can encapsulate non-steroidal anti-inflammatory drugs (NSAIDs) for local use [39] or SPECT tracers [40]. Recently, Fmoc-FF has been utilized in combination with plasmonic gold nanorods (AuNRs) for the development of near-infrared laser-activatable microspheres. These AuNR-embedded dipeptide microspheres, loaded with the anticancer agent, doxorubicin (DOX), were proposed as a smart drug-delivery platform for native, continuous and pulsatile drug release. Results of the study demonstrated the capability to achieve a sustained and on-demand DOX release from the microspheres by manipulating the laser exposure time [41].

Hybrid hydrogel encapsulating docetaxel were also prepared via the calcium-ion-triggered co-assembly of Fmoc-FF peptide and alginate. Due to the synergic effect of these two components, the final material presented much better stability than the single components in both water and a phosphate-buffered solution. Controlled drug release was obtained by varying the concentration ratio between the peptide and the polysaccharide [42]. Fmoc-FF dipeptide has been also utilized to confer mechanical rigidity and stability to natural polymers like hyaluronic acid (HA), a major component of the extracellular matrix. Fmoc-FF/HA composite hydrogels showed a sustained release of curcumin, a hydrophobic polyphenol showing antioxidant, anti-inflammatory, and antitumor activities. Additionally, in this study, it was observed a direct relationship between the rate of curcumin released and the concentration of the Fmoc-FF peptide within the hydrogel matrix [43]. Fmoc-FF/poly-l-lysine (PLL) injectable multicomponent hydrogels, encapsulating the photosensitive drug Chlorin e6 (Ce6), were also proposed as promising delivery platform in the photodynamic antitumor therapy. In vivo studies indicated an efficient inhibition of the tumor growth with no detectable toxicity or damages to normal organs during the treatment [44]. Inspired by Fmoc-FF, recently Adler-Abramovic et al. reported the synthesis of the peptide 6-nitroveratryloxycarbonyl-diphenylylalanine (Nvoc-FF) containing an ultraviolet (UV)-sensitive phototrigger [45]. The UV irradiation of the self-supporting hard hydrogel obtained by the self-assembling of this ultra-short peptide prompts the controlled release of the encapsulated insulin-fluorescein isothiocyanate (insulin-FITC), used as a drug model.

Over the years, multidisciplinary studies have revealed that the self-assembly of short linear peptides and still of single amino acids can make a broad range of diversified materials. It was observed that the conjugation of the polyethylene glycol (PEG) to short homopeptides, containing aromatic amino acids permits a substantial increase of their water solubility. The aromatic residues that can be such are, for example: phenylalanine (Phe, F), tyrosine (Tyr, Y) [46], tryptophan (Trp, W) [47], and naphthylalanine (Nal). Recently, Diaferia et al. proposed a polymer-peptide (PEGylated-F4) functionalized with Gd-DTPA and Gd-DOTA as contrast agent for potential diagnostic applications in magnetic resonance imaging (MRI) [48]. Single peptides in the aggregates are in a β-sheet conformation with an antiparallel alignment along the fiber axis. Each Gd-complex in the nanostructure exhibits a relaxivity value around 15 mM^−1^ s^−1^at 20 MHz, approximately three-fold higher than the classical contrast agents at low molecular weight (4.7 mM^−1^s^−1^). These relaxometric parameters are in line with other examples of Gd(III) based supramolecular (micelles or liposomes) contrast agents [49]. Due to the significant internalization efficiency and due to the high relaxivity values, these nanostructures are able to enhance the MRI cellular response on J774A.1 mouse macrophages cell line. In detail, within those cells the cytotoxicity of the fibril nanoaggregates was negligible with an incubation time of 3 h in the 0.5–5.0 mg/mL concentration range.

The same authors also evaluated the effect of the Gd-complex position in the aromatic framework and of the replacement of the phenylalanine with the non-coded amino acid 2-Nal [50,51]. They observed that the different positioning of the chelating agent into the aromatic framework (at the center or at the *N*-terminus of the F4-motif) causes a drastic loss of the tendency to self-assemble and of the relaxivity value (11 mM^−1^·s^−1^). The decrease of the latter is related to the major flexibility of the Gd-complex on the supramolecular aggregate. On the other hand, the replacement of Phe in the homodimer with non-coded 2-Nal amino acid permits to restore in dipeptide п-п interactions and to prompt self-assembly of Gd-conjugates. This happened thanks to an extended aromatic ring in its side chain. Above 20 mg/mL, Gd-2Nal_2_ peptide derivative gels and it was found able to encapsulate anticancer drugs like DOX, thus suggesting a potential use as theranostic systems.

Recently, Rosenman et al. [21,52] reported the capability of FF and FFF NSs to emit photoluminescence (PL) in the blue and in the green spectral regions, upon thermally-induced reconstructive phase transition. Kaminski et al. [53] also observed the same phenomenon in proteins and in amyloid-like fibrils rich in β-sheet structures. According to the literature, in 2016 Diaferia et al., synthesized series of PEGylated oligo-phenylalanines (PEG_8_-F6, PEG_12_-F6, PEG_18_-F6, PEG_24_-F6) able to generate supramolecular systems rich in β-organization [54,55]. Due to the presence of their β-sheet structures, these polymer-peptides have been proposed as promising bioimaging agents. These peptide nanostructures keep their optoelectronic properties both in solution and at the solid state, upon excitation at 370, 410, and 460 nm, respectively. From a comparison along PEG-series arises that the PEG length and its composition could alter both the structural and the optoelectronic properties of the final material. The differences in the optoelectronic properties were attributed to the extension of the electron delocalization via hydrogen bonds along the cross β-structure of the peptide spine. This effect is due to the number of amide bonds along the PEG chain. With the aim to develop novel biocompatible peptide nanostructures as bioimaging tools, the same authors tried to improve the performance in terms of PL intensity and of wavelength range compatible with in vivo applications. They demonstrated that the intrinsic PL of these peptides nanostructures can be transferred to an acceptor dye like 4-chloro-7-nitrobenzofurazan (NBD) confined in proximity of the nanofiber. Then, the entrapment of NBD in these NSs caused a red-shift from 460 to 530 nm. This evidence was the proof of concept that PL can be red-shifted towards the infrared region of the visible spectrum [56].

### 2.3. Cyclic Peptides

In 1974, theoretical analysis suggested the possible arrangement of a cyclic peptide in a hollow tubular structure [57]. Twenty years later, Ghadiri and coauthors solved the first crystalline structure of nanotube structure, by ring stacking of cyclic peptides incorporating alternating d and l amino acids: cyclo-(l-Gln-d-Ala-l-Glu-d-Ala)_2_ [58]. The peptide side chains were devised on the external area. It is observed that they were organized in the typical nanotube structures, as a consequence of the alternating d and l amino acids. The nanotubes are self-assembled and stabilized by hydrogen bonds between amide groups of the cyclic backbone.

In addition to alternating d- and l-type α-amino acids, several cyclic peptide sequences can make the self-assembly by alternating α- and β-amino acids, β-amino acids, and δ-amino acids by molecular stacking and H-bonds between backbones [59,60,61,62]. The size of the cavity depends on the length of the cyclic peptide, from 2 to 13 Å, increasing from a tetramer to a dodecamer. This parameter with charges on the side chain is essential for the use in biotechnological applications. By tailoring the chemical structure of the cyclic peptide, supramolecular self-assembled architectures can be accustomed to meet the requirements of applications, including stimuli-responsive nanomaterials antibacterial agents, for ion channeling and ion sensing and gene delivery.

Despite a large number of cyclic peptide nanotubes (cPNT) designed, their use as carriers of anti-cancer drugs is very poor. Zhang and co-workers [63] designed an eight-residue cyclic peptide containing Glu and Cys amino acids able to self-organize in a micro-scaled aggregate. PEGylated aggregates loaded with DOX showed a high drug encapsulation ratio. Compared to free DOX, the PEG-modified DOX-loaded CPNT bundles demonstrated higher cytotoxicity, increased DOX uptake and altered intracellular distribution of DOX in human breast cancer MCF-7/ADR cells in vitro.

### 2.4. Amphiphilic Peptides (PAs)

Nature has elected amphiphilic molecules to generate life, by using them to circumscribe portions of the environment. Instead, membranes are able to confine biomolecules and to promote the transport of molecules and ions. Imitating the Nature, amphiphilic peptides self-assemble into different nanostructures, including vesicles, micelles, nanofibers, and nanotubes, thus playing a pivotal role in the production of nanomaterials for biotechnological applications [64,65]. These molecules contain distinct hydrophobic and hydrophilic segments. The most simple peptides able to self-aggregate are constituted by short or long homo-chains of hydrophobic amino acids, like Val, Ala, Gly, and Phe, followed by one or more electrostatic charged residues (such as Asp, Glu, or His). The driving forces of the aggregation are electrostatic (for Asp, Glu, and His residues) and hydrophobic interactions (for Val, Ala, Gly, and Phe residues) that address into a wide variety of nanostructures, depending on their physical and chemical properties.

In 2002, Zhang and co-workers investigated peptide sequences containing an hydrophobic moiety (valine [66], glycine [67] and alanine residues [68]) and an hydrophilic head (one or two Asp residues or one of Lys). These molecules self-assemble into various nanotubes or nanovesicles. In particular, the Lys cationic residues on the head of these peptides could favor their own conjugation with negatively charged DNA and RNA opening the possibility of application in gene drug delivery. Later, Hamley’s group has investigated a cationic peptide in which the hydrophobic tail consists in six Ala residues with an Arg head group. At low concentration, this peptide self-assembles in ultrathin sheets, whereas at higher concentrations the sheets wrap around themself to form nanotubes and helical ribbons. This structure shows antimicrobial properties [69]. It is noteworthy that another sequence able to aggregate is obtained by alternating hydrophobic amino acids with residues bearing on the side chain positive and negative charges. In this case the EAK16 model peptide aggregates into nanofibers [70]. The methyl groups of alanines form a sheet structure inside, whilst the charge residues are exposed on the external wall. This structure is able to delivery ellipticine, an anticancer drug. The UV analysis and the fluorescence demonstrated electrostatic interactions and the conjugation method between the drug and the nanofibers [71].

Aliphatic peptides and lipopeptides were also proposed as building blocks for self-assembling. The feature of lipopeptides is the presence of different short or long alkylic chains as hydrophobic moiety, in the monomer structure. In this case, the aggregation is supported by van der Waals forces. The simplest lipopeptides are also able to form nanostructures. l-dodecanoylserine monomer forms: nanotubes, partially wrapped nanotubes and helical ribbon structures [72]. A peptide amphiphile comprising a single Lys residue, an alpha-(l-Lys),omega-(amino)bolaamphiphile, it was shown to form nanotubes in acidic aqueous solution [73]. Many lipophilic PAs can self-assemble into cylindrical nanofibers, as a consequence of H-bonds among peptide moieties and hydrophobic collapse of alkyl tails [74,75]. The induction of self-assembly in these cylindrical structures could be obtained in aqueous media, in presence of suitable stimuli such as pH [24,40]. The tetrapeptide sequence, composed by hydrophobic and negatively charged residues (Val-Glu-Val-Glu), it grafted to an alkyl tail at sixteen carbon atoms, self-assembled into monodispersed nanobelts in an aqueous solution at a concentration of 0.1 wt% [76]. In this regard, Hartgerink et al. described two different self-assembling modes [74]: acid-induced self-assembly and Ca^2+^ induction. For the acid-induced self-assembly, PA including C_16_H_31_O grafted to C_4_G_3_S(PO_4_)RGD, they can aggregate in nanofibers after dissolution in water and exposition to gaseous HCl. On the other hand, the treatment of a solution of these conjugates with Ca^2+^ instantly caused the gel formation in solution. This Ca^2+^-induced self-assembly may be particularly helpful for medical applications at physiological pH, where formation of a gel is mandatory.

Drug delivery applications of hydrogelation based on PAs were studied by the Stupp’s group [75]. In their studies, the chemical structure of the PA molecule (C_16_V_2_A_2_E_2_) is composed of three segments: an hydrophobic tail (palmytic acid), the well-established β-sheet amino acid sequence V_2_A_2_ and two negatively charged glutamates able to induce cross-linking in presence of Ca^2+^ ions in solution. Properties of this peptide as controlled drug release tool were investigated linking prodan, a fluorescent lipophilic tag used as a dielectric probe for cell membranes. This probe was linked to the peptide through a hydrazine bond inserting a Lys residue at different positions along the backbone of the peptide. This pH-sensitive bond can be broken in acidic conditions of a cell compartment. Hydrogel formation was induced by adding a 100 mM Ca^2+^ solution after dissolving PA in NaOH.

Lipopeptides with similar chemical structure can form self-assembled micelles. Stearyl-H_3_CR_5_C lipopeptide, that are crosslinked by disulfide bonds (SHRss), have been used to form DOX-loaded micelles with an average diameter of 233 nm by nanoprecipitation and probe-based ultrasonication methods. DOX and microRNA are then loaded into the micelles through hydrophobic interactions [77]. The DOX release profile strongly depends on the pH value. Indeed, a larger amount of drug was released at pH 5.5 (82.6%) (corresponding to the endolysomial pH) than at physiological pH 7.4 (63.3%). Cellular uptake has also been investigated by flow cytometry and confocal laser scanning microscopy (CLSM) on DU145 (human prostate cancer) cells. Results indicated that the uptake of micelles were time-dependent with an intracellular uptake rate higher after four hours of incubation (89.50 ± 0.99%) than after one hour of incubation (82.56 ± 1.55%). Cationic micelles can be obtained by self-assembly of PAs in which the hydrophobic moiety is represented by cholesterol (Chol) and the hydrophilic head contains a variable number of positively charged residues, such as histidine and arginine (Chol-CH_5_R_5_, Chol-CH_3_R_3_, Chol-CR_5_, and Chol-CR_3_,) [78]. These aggregates are able to adsorb on their surface different amount of DNA depending on the ratios between the arginine residues and DNA phosphate bases. Formation of micelles with palmitoyl-p53_14-29_has been studied by Missirlis et al. [79]. This PA self-assembled in 10 mM phosphate buffer; the hydrophobic interactions induced the simultaneous formation of micelles with a hydrodynamic diameter of 319 nm and the formation of elongated micelles with a diameter of 10 nm having a length of a few hundred nanometers.

## 3. Self-Assembling PAs for Targeting in Nanostructures

In PA-based nanostructures, the main goal of peptides is to drive the self-aggregation and to regulate the loading and the release of the encapsulated drug. In the wide category of PAs, the sequence of the amino acids is responsible for the targeting and delivery features. For this purpose, the sequence is selected on the ability to cross the cell membrane or to bind overexpressed receptors on the cell membranes.

### 3.1. Cell Penetrating Peptide (CPPs) and Smart Sequences

Cell penetrating peptides (CPPs) are a large class containing more than 1700 different experimentally-validated sequences [80,81]. Most common CPPs are cationic and they are widely used. A class of CPPs are derived from the α-helical domain of the Tat protein, covering residues from 48 to 60. Those residues are mainly composed of basic amino acids, such asthe TAT dodecapeptide: GRKKRRQRRRPQ [82]. The CPP role as a nanovector has been described in many reviews [83,84].

Despite high cellular uptake efficiency, CPPs lack cancer cell specificity. To overcome this drawback stimuli-responsive CPPs have been developed recently to enhance the cellular uptake of therapeutic cargo only in the tumor tissue. As previously reported, these stimuli can respond to pH variation or to enzyme activity or to oxidative stress. CPPs can be considered as responsive molecules when containing residues able to vary the net charge depending on the pH. One residue able to tune the net charge of the peptide is His. Therefore, Zhang et al. designed an α-helical CPP to obtain a pH-responsive peptide, by replacing all its lysines with histidines (THAGYLLGHINLHHLAHL(Aib)HHIL). This peptide (TH) showed a neutral charge at physiological pH, but the net charge became positive under acidic conditions, so that its cell penetration capacity was activated [85]. PEGylated liposomes functionalized with TH peptide showed a more efficient internalization into C26 colon cancer cells, when compared to non-targeted liposomes. Moreover, PTX-loaded liposomes suppressed C26 colon tumors in vivo with high apoptosis levels where the tumor inhibition rate reached 86.3%. Proteases abundant in tumor tissue can constitute internal stimuli for activating CPPs. Liu et al. developed a liposome able to carry DOX labelled with the sequence AAN-Tat [86]. The AAN sequence is a substrate of Legumain endoprotease. The addition of the AAN moiety to the fourth lysine in the TAT generates a branched peptide moiety, which leads to a decrease in the transmembrane transport capacity of TAT up to 72.65%. The action of the enzyme allows restoring the penetrating capacity of TAT. In vivo assays carried out on nude mice showed inhibition of the tumor growth significantly higher in mice administered with AAN-TAT liposomal DOX than control. Simultaneously, for the group treated with targeted DOC liposomes, it was also observed a prolonged survival period. Another stimulus could be derived externally, such as UV or NIR irradiation. In these cases, the CPP peptide is modified by the action of a residue with a photolabile protecting group. Upon reaching the tumor site, the peptide-functionalized liposomes are irradiated by UV or NIR light that cleaves the protective groups and in this way allows the CPP to play its role. Following this methodology, a CPP (CKRRMKWKK or CGRRMKWKK) enhances the efficiency of the translocation, derived from the penetration. This CPP is designed to turn into an inactive form by neutralizing the positive changes of the lysines, which are caged with photoresponsive groups [87].

### 3.2. Peptide Able to Interact with Overexpressed Receptors

In certain PAs, the receptor-targeting peptides are able to induce high levels of internalization within tumor cells due to a receptor-mediated endocytosis mechanism. The peptide sequence can be composed in this manner [88]. These strategies could allow the intracellular delivery of the payload. Some known endogenous proteins are able to bind the target receptor with high affinity. A significant topic of research is about how to preserve the affinity for the overexpressed receptors, especially after the conjugation to the hydrophobic moiety. Furthermore, evidence showed how all the residues which are involved in the receptor binding are well-exposed on the nanostructure surface. Those residues maintained a conformation suitable to the interaction with the receptor [89]. Further studies are aimed to preserve the in vivo chemical stability, due to the high sensitivity of peptides to the protease degradation. Improved metabolic stability and pharmacokinetics can be achieved by modifying peptide sequences with either specific coded amino acids or un-coded amino acids, or, as well, with amino acids in the d configuration. Alternative strategies consist of: the cyclization between the *N*- and *C*-terminals, the cyclization between the *N*- or *C*-terminal and a side-chain, or the cyclization between two side-chains.

Peptide sequences can act as cell surface receptor antagonists if molecules are modeled, allowing selective targeting towards to receptors. Antagonist peptides show a dual advantage if compared with their agonist counterparts: on the one hand they do not act in the biological pathways following receptor binding; on the other hand they present higher binding capacities [90,91]. However, the strategy of rational design of new compounds has some limits, one of the most significant being the limit of the knowledge requirement related to the structure of ligand/receptor interaction [92]. A further possibility to identify novel peptide sequences is the use of the phage display technique, concerning recognizing tumor-associated proteins [93]. Next to the identification of a peptide sequence, some suitable spacers (charged or neutral) can be inserted between the hydrophobic region and the peptide. One of the most used spacers in this context is uncharged polyethylene glycol (PEG). Indeed, the presence of one or more ethoxilic units permits an increase in the blood circulation time of the supramolecular aggregates (enhanced permeability retention effect). In addition, the lack of charge prevents interactions with the residues on the bioactive portion which could induce unnatural and incorrect conformations. Furthermore, the spacer allows the maintenance of the molecule’s flexibility, mobility, and increases, in some cases, the solubility. The formulation of supramolecular aggregates, like micelles and liposomes, externally functionalized with bioactive peptides, may be obtained by using different approaches, such as pre-functionalization and the post-functionalization strategies [94,95,96,97]. In the pre-funtionalization, the peptide sequence is placed on the aggregate during the nanostructure preparation: Figure 3 (left panel) shows an amphiphilic peptide derivative added during the formulation step. In the post-functionalization strategy (Figure 3, right panel), the peptide is chemically conjugated on the aggregate surface after nanostructure organization. The advantages of the first method are a defined quantity of bioactive molecules in the aggregate and the avoidance of impurities, but it requires as input a well-purified amphiphilic peptide molecule. However, in the case of liposomes, the bioactive peptide is located on both the external surface and in the inner aqueous compartment, after liposomial formulation. In the post-functionalization approach, peptide coupling concerns the introduction of suitable functional groups (already activated) onto the external side of liposomes or nanoparticles for covalent or non-covalent peptide binding. In order to make sure about the proper orientation of the targeting ligand, biorthogonal, and site-specific surface, it is necessary to choose the appropriate reactions. In this sense, the most used chemical approaches are: enzymatic ligation, Cu-free chemistry, the amine in case of the *N*-Hydroxysuccinimide coupling method; thiol for maleimide; Michael addition; azide for Cu(I)-catalyzed Huisgen cycloaddition (CuAAC); biotin for non-covalent interaction with avidin;triphosphines for Staudinger ligation; and hydroxylamine for the oxime bond [98].

In the literature, targeting peptides are tailored toward three broad types of receptors which are overexpressed or exclusively expressed in cancer vasculature and/or cancer cells: integrins; growth factor receptors (GFRs); and G-protein-coupled receptors (GPCRs). Several examples of supramolecular systems loaded with therapeutic or diagnostic agents and externally decorated with homing peptides, able to selective recognize integrin receptors or membrane receptors belonging to the GPCR superfamily, are reported in Table 1.

#### 3.2.1. Peptide Target for Integrin Receptors

Integrins are heterodimers transmembrane receptors related to the cell-extracellular matrix (ECM) adhesion. Upon ligand binding, integrins activate cellular signals such as regulation of the cell cycle, organization of the intracellular cytoskeleton, and movement of new receptors to the cell membrane. Integrins are one of the most important receptors that can be used in active targeting strategies [122]. Among the different subfamilies of these heterodimeric transmembrane proteins, integrins α_V_β3 and α_V_β5 have prominent roles in angiogenesis and metastatic disseminations. The integrin αvβ3 plays a very domineering role in angiogenesis and is overexpressed in endothelial cells of the tumour. Recently a large exploration in the field of αvβ3 integrin-mediated bioactive targeting for cancer treatment has been reported. All designed peptide sequences contain the RGD motif.

In most of the cases, the cyclization is commonly employed to improve the binding properties, conferring rigidity to the structure. In linear peptides, the fourth amino acid alters the binding specificity and the nature of residues, by flanking the RGD sequence. The fourth amino acid could influence receptor affinity, receptor selectivity, and other biological properties [123]. Therefore, nanoaggregates grafted with cRGD sequence have been widely evaluated for the treatment of different cancers, such as ovarian cancer, melanoma, and breast carcinoma [124,125]. The first examples of aggregates functionalized with RGD containing peptides they were formulated only in half of the last decade. In fifteen years, more than 450 articles were published on RGD-labelled liposomes or micelles delivering hydrophilic drugs like DOX [126,127], but also with many others anticancer drugs, such as cisplatin (CDDP) [128], paclitaxel (PXL) [129,130], docetaxel (DTX) [131], combretastatin A4 (CA4) [132], and 5-fluorouracil (5-FU) [133,134].

Generally, the RGD sequence is inserted in a five residue cycle in which one of the amino acids is in the d-configuration. Therefore, in the case of a single d-amino acid and four l-amino acids, the homodetic cyclic pentapeptide prefers a II’/conformation with the d amino acid in the i + 1 position of the II′-turn [135]. Most aggregates were grafted with the c(RGDfK) cyclic peptide (Figure 4a). This sequence, developed by Kessler et al. [136], is able to target the αvβ3 and αvβ5 integrin receptors [137]. For example, c(RGDfK) was coupled to poly(l-lactide)-block-poly(ethylene glycol)-succinic ester (NHS-PEG-PLA) [138] to obtain polymeric micelles able to deliver hydrophobic drugs like CA4 [99].

The density of cRGD on the micelle surface can affect the amount of drug delivered into the cells. This issue was studied by Kataoka’s group varying the quantity the monomer functionalized by cRGD ligand from 5% up to 40%. cRGDfK-labelled micelles were prepared by “post conjugating” Cys-containing cRGD peptides onto maleimide-functionalized DACHPt/micelles obtained from a mixture of poly(ethylene glycol)-*b*-poly(l-glutamic acid) (MeO-PEG-*b*-P(Glu)) and maleimide-conjugated poly(ethylene glycol)-*b*-poly-(l-glutamic acid) (Mal-PEG-*b*-P(Glu)) [100]. The better results in terms of uptake and cytotoxicity were observed for cis-platinum-loaded micelles functionalized with an amount of peptide ranging between 20 and 40%.

Others analogue cyclic RGDs were also analysed for drug delivery applications. One of those studies was performed by Tao et al., who formulated DOX loaded liposomes labelled with c(RGDfC): the peptide was conjugated to the liposomal surface by a thiol-maleimide coupling reaction with MBPE lipid [MBPE-c(RGDfC)] and the PEG coating of liposomes was obtained by using the post-insertion method [101]. Key factors in tumor therapy are biodistribution and clearance of the aggregates affected by peptide hydrophilicity. In view to assess the effect of the peptide hydrophilicity, PEGylated liposomal DOX has been tested in vitro on integrin-expressing HUVEC cells, after the insertion of three cyclic RGD analogues: RGDyC, RGDf[N-Met]K, and RGDfK. Liposomal systems grafted with the RGDf[N-Met]K sequence were than compared to the other two analogue sequences (RGDyC and RGDfK). They showed the lowest undesired (not) specific interactions with other integrin-presenting sites, localization in tumor, and lower DOX side effects [102]. Moreover, a further cyclic sequence c(RGDyk) was then put in place as a labelling cisplatin delivery system for therapeutic applications against bone metastasis derived from prostate cancer in a mouse model [103]. CDDP-loaded targeted liposomes showed a higher cytotoxicity (IC_50_ = 1.83 µM) than free drug (IC_50_ = 15.4 µM) or untargeted liposomal drug (IC_50_ = 10.0 µM). The capability of these targeted liposomes to be selectively accumulated in metastatic tumor bones was tested during several in vivo assays. They showed a clear tumor regression.

Moreover, other analogues were obtained introducing in the peptide cycle of the cyclo azabicycloalkane and aminoproline residues. Zanardi et al. arranged targeted liposomal doxorubicin by incorporating a 5% molar ratio of DSPE-PEGcAbaRGD or DSPE-PEG-cAmpRGD amphiphiles into cAbaRGD-LP or cAmpRGD-LP, respectively [104]. They also studied their in vitro behaviour on three different cell lines (MCF7, HUVECs, and HepG2). Results showed how both targeted liposomes (cAbaRGD-LP or cAmpRGD-LP) possess higher kinetics of nuclei internalization and a higher percentage of cell death when compared to the free drug.

In the last decade, a new cyclic peptide (CRGDKGPDC) iRGD was identified for peptides hosting tumor metastases [139]. This peptide was found to bind αβ integrin overexpressed on the surface of cancer cells and on tumor-vessel cells, but not in normal vessel cells. In delivery applications, the peptide was anchored on the surface by a post-insertion method, in turn, to develop the iRGD properties of several aggregates transporting the isoliquiritigenin (ISL), a natural anti-breast cancer dietary compound [140] or the doxorubicinand sorafenib [105]. However, self-assembling iRGD-based amphiphilic molecules have rarely been reported. The targeting motif was chemically modified with a hydrophilic arginine-rich sequence and hydrophobic alkyl chains sequentially able to self-assemble in a nanostructure. This adjustment aimed to deliver photosensitizer hypocrellin B for photodynamic application the iRGD [141]. Moreover, very recently, a new PA containing iRGD and a hydrocarbon chain, in addition to hydroxyethyl starch (HES), a semi-synthetic polysaccharide (iRGD-HES-SS-C18 NCs), was formulated [142].

#### 3.2.2. GPR Target Peptide

A wide number of nanostructures were functionalized with peptides able to recognize GPCRs, in particular to target receptors for somatostatin (SST), cholecystokinin (CCK), gastrin-releasing peptides (GRP/Bombesin), lutein, and neurotensin.

##### Somatostatin Receptors

Nanoaggregates directed toward somatostatin receptors have been widely exploited for diagnostic and therapeutic applications. Instead, a side effect is highly frequent in the expression of SSTR in human tumors of neuroendocrine origin, mostly affecting the expression in normal tissues. In general, SSTR2 is the most common SSTR subtype found in human tumors, followed by SSTR1, with SSTR3, that are four and five times less common. Due to the very low in vivo half-life of the wild-type SST tetradecapeptide, researchers have preferred to label aggregates with more stable somatostatin analogues. The most renowned selected analogue is the octreotide (OCT) shown in Figure 4b. The OCT is a cyclic peptide containing eight amino acids in l and d configuration, developed in 1992 by Sandoz (now Novartis) [143]. This cyclic peptide is able to cross the cell membranes via endocytosis by binding to SSTR2 with high affinity and inhibitory concentration better than wild-type SST (IC_50_ = 2nM). The OCT is able to bind also SSTR3 (IC_50_ = 376 nM) and SSTR5 (IC_50_ = 299 nM), but with a lower degree if compared to SSTR2. Moreover, the OCT peptide binding to receptors is not affected by chemical modifications on its *N*-terminus.

Octreotide-labeled aggregates may be obtained following both previously reported approaches and employing opportune strategies aimed to avoid possible undesired compounds. Many studies demostrated that the β-like turn, formed by Phe-Lys-DTrp-Thr residues, is involved in receptor binding [144]. Therefore, it is essential to verify the retention of the amino acid configuration and the exposition of the tryptophan residue on the external aggregate surface. Hence, after their formulation, peptide properties on the liposomes have to be fully characterized. In 2009, Morisco et al. synthetized OCT amphiphlic molecules able to self-assemble in micelles for the selective delivery of magnetic resonance imaging (MRI) contrast agents [106]. These PAs contain three different regions: a hydrophobic moiety based on two stearyl chains, a chelating agent (DTPAGlu or DOTA) able to coordinate Gd^3+^ ions as a contrast agent, and the bioactive peptide. Fluorescence studies indicate for all micelles a complete exposure of OCT on the surface. CD measurements show the predominant presence of a β-sheet peptide conformation, characterized by a β-like turn.

The majority of aggregates in the literature are not formulated only by PA self-assembling, but they are obtained by mixing PAs with other surfactant molecules. Morelli’s group has studied mixed aggregates formulated by co-assembling of the OCT lipopeptide with a second monomer containing in the hydrophilic head: a metal complex acting as diagnostic or therapeutic agent. In diagnostic aggregates (Gd-DTPAGlu, Gd-DTPA, and Gd-DOTA complexes), the metal chelate is covalently bound through a lysine residue to two eighteen-carbon chains [107]. Structural characterization of the aggregates indicates a shape and size of the supramolecular aggregates suitable for in vivo use. Therapeutic aggregates were formulated by co-assembling, at a 10/90 molar ratio, of the OCT lipopeptide with a second amphiphilic monomer containing a cytotoxic platinum complex anchored to the lipophilic tails, (C18)_2_PKAG-Pt [108]. The (C18)_2_-PKAG-Pt/(C18)_2_(AdOO)_5_-OCT mixed aggregates generate large liposomes with an average diameter of 168 nm. These liposomes were further loaded in their inner aqueous compartment with the hydrophilic DOX drug. Indeed, platinum complexes are frequently used as chemotherapeutics, in combination with other drugs such as paclitaxel, bleomycin, vinblastine, and in several trials with DOX. This represents the proof of concept of combined therapy based on DOX and platinum complexes.

Targeted OCT aggregates were also largely investigated as carriers for the delivery of hydrophobic anticancer drugs, such as paclitaxel (PTX): a mitotic inhibitor used to treat patients with lung, ovarian, breast, head and neck cancers, and advanced forms of Kaposi’s sarcoma. Zhang et al., loaded PTX in polyethylene glycol-polycaprolactone (PEG-PCL) polymeric micelles, they obtained the OCT-(PTX)-PEG-b-PCL (OCT-M-PTX) and the salinomycin (SAL)-loaded PEG-b-PCL (M-SAL). The OCT was coupled to NHS-PEG-b-PCL through the activated NHS group [109]. These micelles had a diameter of approximately 25–30 nm and the encapsulation efficiency of the drug was 90%. Moreover, by adding free OCT, the interaction was inhibited, then it was confirmed that cellular uptake occurs through a receptor-mediated mechanism.

Zou et al. coupled OCT to hydrophobilized chitosan polymer [110]. This peptide derivative was able to self-assemble in micelles having very low cytotoxicity, an excellent biocompatibility, and biodegradability. In detail, the authors formulated *N*-octyl-*O*,*N*-carboxymethyl chitosan (OCC) and *N*-deoxycholic acid-*O*,*N*-hydroxyethylation chitosan (DAHC) micelles. Then, they conjugated the OCT on the *N*-terminal moiety of free carboxylic groups of OCC. The coupling had an extremely low (about 3%) yield, which is largely due to the high molecular weights of OCT and chitosan derivatives, due to the strong hydrogen bonds in the chitosan backbone, and due to poor solubility of chitosan derivatives in organic solvent. This result pushed toward alternative mixed aggregates, adding to DAHC a ligand-PEG-lipid conjugate able to guarantee same long circulation time in blood and ligand targeting. Both micelle types showed good DOX loading capability, with a drug loading content (DLC) in the 22–30% range. As an alternative strategy, the same authors anchored *N*-terminal peptides in solution to a PEG fragment and this moiety was conjugated to an aliphatic chain or to the deoxycholic acid obtaining the OCT(Phe)-PEG-SA (OPS) monomer or the OCT(Phe)PEG-DOCA (OPD), respectively [110].

More recently, OCT-functionalized unimolecular micelles were exploited to delivery thailandepsin-A (TDP-A) toward neuroendocrinal tumor cells. TDP-A is a relatively new naturally produced histone deacetylase (HDAC) inhibitor. Target selective micelles were obtained by the self-assembling of individual hyperbranched polymer molecules, consisting of a hyperbranched polymer core (Boltorn® H40) and approximately 25 amphiphilic polylactide-poly(-ethlyene glycol) (PLA-PEG) block copolymer arms covalently bound through the succinimidyl group (NHS) to octreotide (H40-PLA-PEG-OCT) b [111].

Helbok et al. [112] synthesized an amphiphilic OCT derivative by cross-linking the *S*-acetyl-mercaptopropionic acid peptide (SAMA-TOC) with the Mal-DSPE-PEG2000 phospholipid. Next mixed liposomes were obtained by adding to the OCT derivative adequate amounts of palmitoyloleoyl-phosphatidylcholine (POPC), lyso-stearyl-phosphatidylglycerol (Lyso-PG), distearyl phosphatidylcholine–polyethyleneglycol-2000 (DSPE-PEG2000), and dimyristoylphosphoethanolamine-DTPA (DMPE-DTPA) in a molar ratio of 0.1:11:7.5:0.9:2, respectively. These aggregates are usually employed in nuclear medicine applications radiolabelling with indium-111.

Octreotide-targeted liposomal doxorubicin was constructed with different ligand density by post-inserting HSPE-PEG4000-Octreotide into pre-formed liposomes. The octreotide ligand insertion was confirmed by the activity detection of octreotide in HSPE-PEG4000-Octreotide with synchronous fluorescence. Results indicated that an octreotide density around 1% could achieve the best uptake efficiencyon NCI-H-446 and SMMC-7721 cell lines among the studied liposomes [113].

Similar properties were shown by the somatostatin [Tyr3]-octreotate (TATE) analogue. Petersenet al. [114] conjugated this peptide to maleimide, covalently attached to the distal end of DSPE-PEG2000 via a thioether bond. Targeted liposomes (DSPC/Chol/DSPE-PEG2000/DSPE-PEG2000-TATE in a molar ratio of 50:40:9:1, respectively), they encapsulated a positron emitter ^64^Cu, as diagnostic agent for positron emission tomography (PET) imaging. Peptide-labelled liposomes displayed significantly higher tumor-to-muscle (T/M) ratio (12.7 ± 1.0) compared to control-liposomes without TATE (8.9 ± 0.9) and to the ^64^CuDOTA-TATE peptide (7.2 ± 0.3). These results reveal the feasibility of utilizing somatostatin analogs for specific targeting of the above-described aggregates to tumors overexpressing somatostatin receptors.

Very recently, TDP-A and AB3, new histone deacetylase inhibitors, they were encapsulated in the hydrophobic core of self-assembling micelles labelled with a somatostatin analog KE108 (PAMAM–PVL–PEG–OCH3/Cy5/KE108) [115,116]. This nonapeptide analogue contains the Phe–d-Trp–Lys–Thr motif, crucial for high-affinity somatostatin receptor binding like octreotide. Being formed by eight residues, the cycle size of this analog is larger than octreotide. It possesses high affinity to all five subtypes of SSTR. KE108 exhibited superior targeting ability in medullary thyroid cancer (MTC) cells, if compared to octreotide.

##### Bombesin Receptors

The four receptor subtypes which are associated with the Bombesin-like peptides (BLP) family have been identified and found to be overexpressed in prostate, breast, small cell lung, [145] ovarian, and gastrointestinal stromal tumors [146]. A peptide able to bind these receptors is the bombesin (BN), which is constituted by fourteen aminoacid residues. Its eight-residue *C*-terminal peptide sequence ([7–14]BN), reported in Figure 4c, and many other BN analogs have been modified to selectively carry diagnostic or therapeutic agents to their receptors. They act both as agonists or antagonists. Many studies demonstrate that the [7–14]BN fragment and its analogues conjugated on the N-terminus with amino acid linkers, aliphatic or hydrophilic moiety, they all keep the affinity for receptors [147,148,149].

Despite the large overexpression of these receptors only few aggregates were developed. In the first example, Accardo et al. prepared mixed liposomes composed by two amphiphilic derivatives (C18)_2_-L5-[7–14]BN (or (C18)_2_-PEG3000-[7–14]BN) and (C18)_2_-DOTA(^111^In), both of them containing the same hydrophobic portion (two stearyl tails) and alternatively BN peptide or indium complex. The presence of a metal complex could allow to detect the in vitro fate of the liposome and its binding capability. Peptide was anchored to the alkyl chains trough different ethoxylic spacers (L5 or PEG3000). These spacers permit to improve the hydrophilicity of the final monomer and to increase the bioavalability of the peptide sequence on the external surface of the liposome. It is worth noting how to perform a really active targeting, as it is relevant to consider the length of the ethoxylic region. In fact, a long chain hides the bioactive sequence [117]. Successively, the same authors synthetized DSPC-based liposomes derivatized by the pre-functionalization approach with the MonY-BNAA1 monomer containing [7–14]BN analogue, DOTA chelating agent and the alkyl chains in the same molecule. Specific binding capability and cytotoxicity of these targeted liposomes, loaded with DOX, were carried out in PC-3 xenograft-bearing mice. An inhibition of the tumor growth in mice treated with DSPC/MonY-BN/DOX targeted liposomes [118] was observed.

More recently, the same sequence was grafted to cholesterol by a click chemistry, following a post-insertion method. The liposome obtained by mixing this monomer with DPPC was able to load subphthalocyanines (SubPc), an interesting hydrophobic probe for optical imaging, with a geometry that prevent aggregation [150]. An amphiphilic derivative of the [7–14]BN peptide was also used to prepare sterically-stabilized mixed micelles (SSMMs) as drug delivery systems for gold(III) complexes (AUL12). The latter is already known for its in vitro and in vivo high antitumor activity, even in the CDDP-resistant cell lines. These micelles were able to encapsulate the hydrophobic metal complex with high loading efficiency while maintaining the gold (III) complexs table in the +3 oxidation state over a period of 72 h. The in vitro binding ability and cytotoxicity of this target selective micelles were assessed in PC-3 cells overexpressing the GRP/bombesin receptors [119].

Anyway, circulation time in vivo of the [7–14]BN wild-type (t_1/2_ = 15.5 h) remains relatively short. This evidence led the same authors to develop a new peptide analog, BNAA1, in which Sta^13^-Leu^14^ and the Gly^11^ residue with the *N*-methyl-glycine replaced Leu^13^-Met^14^ residues. These changes were finalized to increase the resistance towards the aminopeptidase and carnitine enzymes. The labelled DSPC/MonY-BN-AA1/DOX liposomes reduce the tumor volume showing value reductions superior to 20% when compared toDSPC/MonY-BN/DOX liposomes [120].

##### CCK receptors

In neuroendocrine origin tumors, such as medullary thyroid cancers, it was found that both CCK1 and CCK2 receptors were overexpressed. The same phenomenon was found in small cell lung cancers and in gastroenteropancreatic (GEP) tumors. The peptide CCK8 is able to recognize both receptors.

In Figure 4d, one can see the eight residue C-terminus sequence of the endogenous hormone cholecystokinin (CCK). The CCK8 can be tailored on *N*-terminus without affecting receptor binding. This feature is essentially due to the interaction of receptor *N*-terminal extra domain with the amino acid side chains. The latter lies on the *C*-terminal moiety of the peptide ligand, as demonstrated by solution NMR [151] and theoretical studies [152]. Based on these data, in the last 10 years, Accardo et al. developed a wide class of CCK8-decorated supramolecular aggregates (namely Naposomes), by anchoring the bioactive moiety through the *N*-terminus [107]. The exposition of the CCK8 peptide was assessed by fluorescence measurements [153]. However, the peptide availability on surface aggregates is not an exclusive requirement for the receptor binding: the correct peptide conformation is crucial to assure high affinity and selectivity in ligand/protein binding processes. In this case, the CCK8 peptide needs to adopt a pseudo-α-helix conformation to give high binding affinity towards to the CCK1-R and CCK2-R receptors, according to the membrane-bound pathway theory [151]. The authors demonstrated that only peptide amphiphiles having an initial random coil conformation were able to adopt the pseudo-α-helix conformation in the presence of the receptor. Unlike them, peptides like (C18)_2_-L5CCK8, in which the peptide displays a β−sheet conformation, do not show in vitro cellular binding. Closing that chemical modification on the CCK8, the *N*-terminus seems to play an important role in stabilizing the peptide active conformation in self-assembling. The CCK8 amphiphilic monomers were combined with a second monomer containing the DOTA or DTPA chelating agent (general formula: (C18)_2_-L5CCK8 and (C18)_2_-CA, respectively). The morphology and the size of the resulting aggregates (micelles, liposomes or open bilayers) are determined by several parameters, such as ionic strength, pH, monomer structure (length of polioxiethylene spacers), composition, and formulation procedure (dissolution in buffered solution or well-assessed procedures based on sonication and extrusion) [107,121,154]. Moreover, these aggregates can play a double role as theranostic delivering contrast agents and drugs [155].

#### 3.2.3. Supramolecular System Based on Disordered Linear Peptides

The design of supramolecular systems could drive the disordered peptides to fold into a stable structure. This structural modification could be a promising route to develop a new class of bio-molecules for processes in which a specific conformational rearrangement is required [156]. These considerations deserve an in-depth study of the intrinsic disorder of peptide behavior in solution and their performance on surface of nanostructures [157]. Recently, the authors have studied the structural preferences of linear synthetic peptides with CPC-containing sequences (chemokine receptor CXCR4) characterized by the presence of some unordered amino acids [158]. In particular, these studies showed the conformational flexibility of both peptides, tested on the CXCR4 receptor through an indirect binding assay. Additionally, the authors tested the inhibition of CXCL12-induced migration and cAMP reduction. In addition, they proved how disordered peptides possess a stronger inhibitory capability on the adenilate cyclase, if compared to the AMD3100, which is, nowadays, the best characterized CXCR4 inhibitor. Trial evidence highlights that short, flexible peptides with no regular secondary structure can dynamically explore some conformational ensembles by targeting the chemokine receptor CXCR4. The employment of intrinsically-disordered peptides could lie in the skill to control the transition between different structural states, especially as biosensors and in molecular recognition [159].

## 4. Conclusions

In this review, we have reported the most recent evidence on peptide-based drug-delivery systems in biotechnological applications. During the examination of the very rich literature data, several very remarkable and significant aspects have come into sight. Without a doubt, the extensive use of peptides to build more complex molecular constructs for biotechnological applications is well known. This is mostly due to their ease of achieving, and automation in, the synthesis of ad hoc designed sequences. In addition, peptides are also suitable for modification and control to gain desired biostructures in different aggregates. Additionally, several of their specific features allow operative research groups to obtain a broad variety of biotechnological materials. Furthermore, the option to arrange them in both linear and cyclic peptide sequences is worth mentioning; the likelihood to draw on side chains of the amino acid residues; the possibility to load on them charges and functional groups; and, finally, the intrinsic opportunity to arrange predictable physical and chemical patterns, suitable for biotechnological modular applications. In structured and/or disordered peptides, we can also consider the option of using conformational preferences: we can always put up micelles, liposomes, and gels based on peptides with preferential secondary dimensions and structures.

All these characteristics can also be engaged by means of bioactive sequences and/or through the recognition of post-translation moieties between biosystems. Therefore, it seems evident that the concrete possibilities that these biomaterials open up many sectors of peptide research, which can be engineered for specific applications in the various biotechnology sectors.

As said above, nature itself has elected amphiphilic molecules to generate life, by using them to circumscribe a portion of the environment. Indeed, surfactants constitute membranes able to contain biomolecules inside cells and they can select and transport molecules and ions. While imitating nature, amphiphilic peptides self-assemble into different nanostructures, such as vesicles, micelles, nanofibers, and nanotubes. In our guess, this is the way can play a key role in the production of new nanomaterials designed for biotechnological applications.

## Figures and Tables

**Figure 1 molecules-24-00351-f001:**
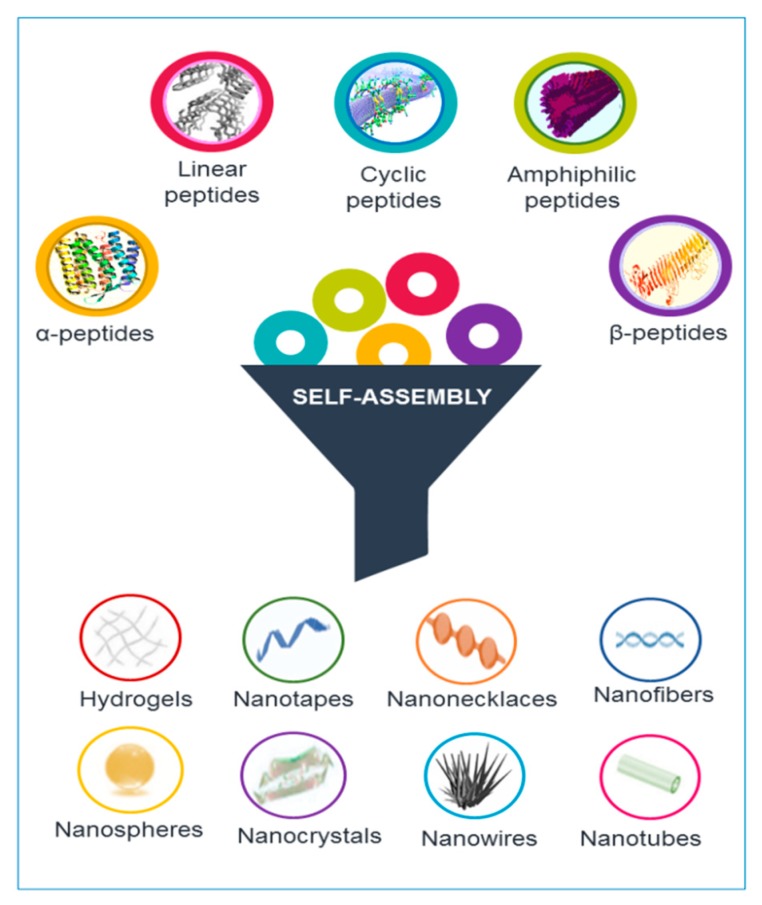
Different classes of peptides can be arrange in supramolecular structures handling the self-assembling phenomena. Various morphologies can be generated according to the rational design of the primary sequence.

**Figure 2 molecules-24-00351-f002:**
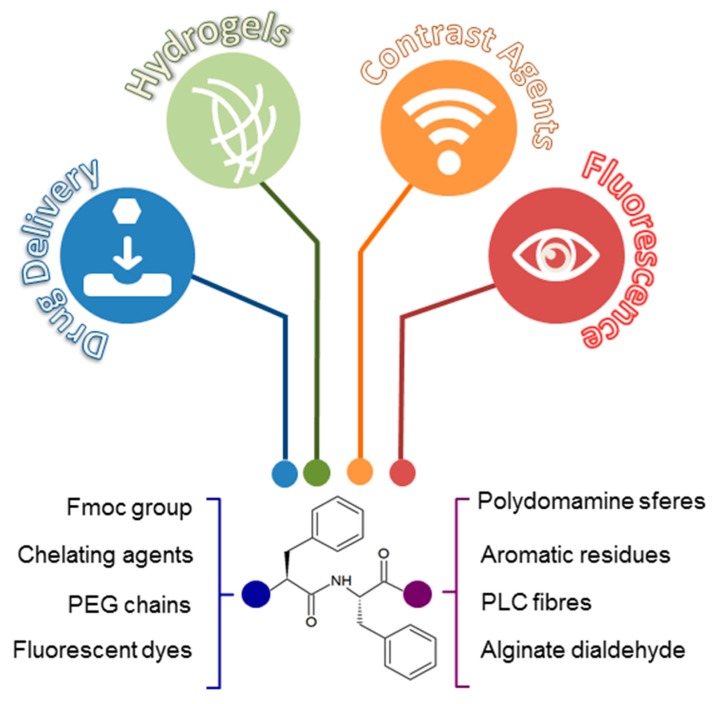
Diphenylalanine based aggregates can be applied with different biotechnological scopes, producing drug delivery systems, hydrogels matrices, supramolecular contrast agents, and fluorescent aggregates. Chemical and functional decorations (like sequences modification, incorporation of fluorescent dyes and conjugation with chelating agents and polymers) at *N*- and *C*-terminus of the primary sequence produce innovative nanostructurated tools.

**Figure 3 molecules-24-00351-f003:**
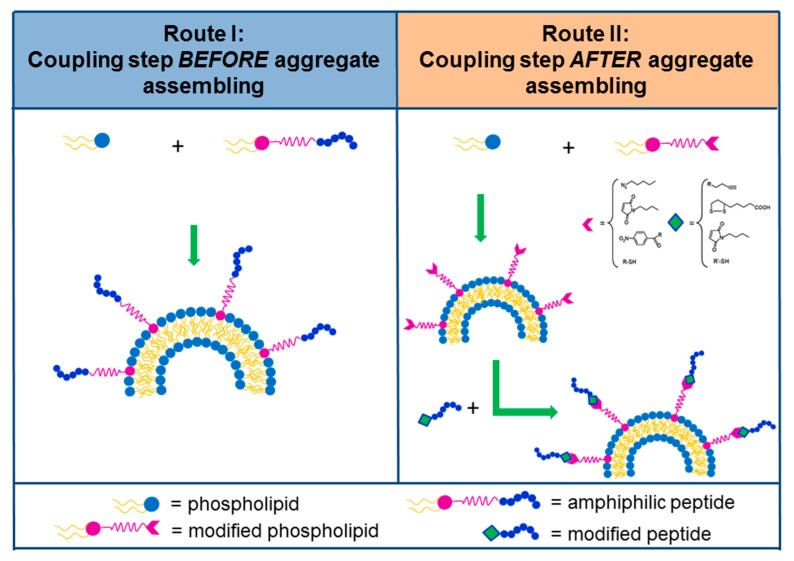
Schematic representation of two approaches used for the synthesis of peptide containing liposomes. In route I (pre-liposomal functionalization) a PA is inserted directly during the liposome formulation. In route II (post-liposomal functionalization) a peptide is anchored on the exteral surfaces after liposome formulation by a selective reaction between two functional groups displayed on the peptide and on the liposome, respectively.

**Figure 4 molecules-24-00351-f004:**
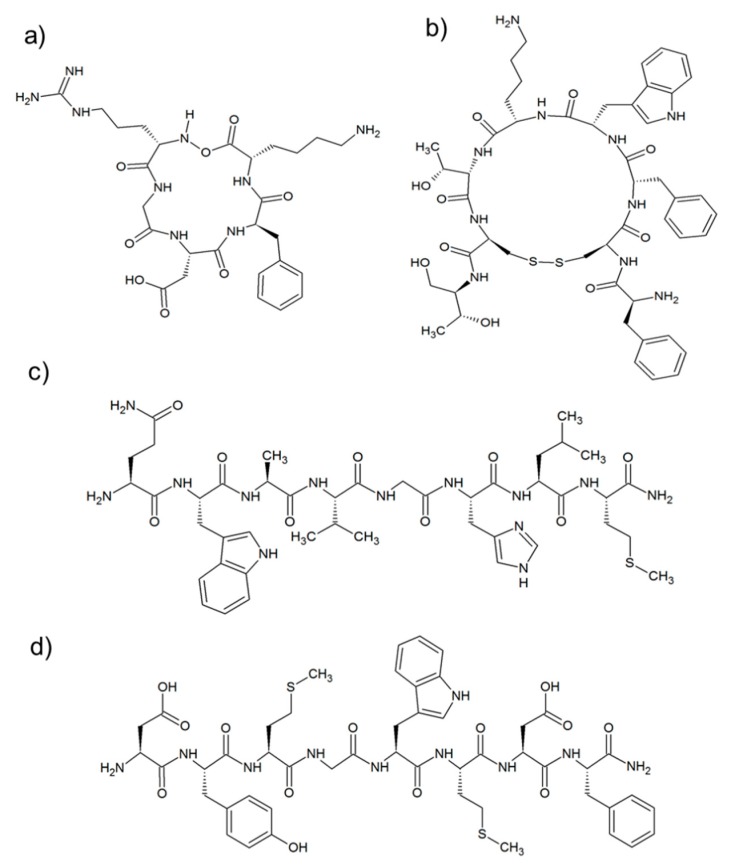
Structure of targeting peptide sequences: (**a**) c(RGDfK); (**b**) Otreotide[c(2–7)-FCFWKTCTol]; (**c**) [7–14]-Bombesin (H-QWAVGHLM-NH_2_), and (**d**) CCK8(H-DYMGWMDF-NH_2_).

**Table 1 molecules-24-00351-t001:** Supramolecular systems decorated with homing peptides able to selectively recognize integrin receptors or membrane receptors belonging to the GPCR superfamily. The bioactive peptide, the peptide conjugation, the encapsulated API, and the corresponding references are reported.

Receptor	Peptide Sequence	Peptide Derivative	Drug	Ref.
Integrin receptor Avβ3	c(RGDfK)	c(RGDfK)-NHS-PEG-PLA	CA4	[99]
	c(RGDfK)	c(RGDfK)-SH post liposome modification	CDDP	[100]
	c(RGDfC)	MBPE-c(RGDfC) post-insertion	DOX	[101]
	c(RGDf[N-Met]K)	c(RGDf[N-Met]K(Ac-SCH2CO))	DOX	[102]
	c(RGDyK)	DSPE-PEG- c(RGDyK)	CDDP	[103]
	cAbaRGD cAmpRGD	DSPE-PEG-cAbaRGD DSPE-PEG- cAmpRGD	DOX DOX	[104]
	iRGD	iRGD-HES-SS-C18 NCs	DOX/sorafenib	[105]
G-Protein coupled receptor	Octreotide	OCA-DOTA/ OCA-DTPAGlu	Gd-complex	[106]
	Octreotide	(C18)2(AdOO)5OCT	Gd-complex	[107]
	Octreotide	(C18)2(AdOO)5OCT	CDDP/DOX	[108]
	Octreotide	OCT-(PTX)-PEG-b-PCL	PTX	[109]
	Octreotide	Oct-Phe-PEG-SA	DOX	[110]
	Octreotide	H40-PLA-PEG-OCT	TDP-A	[111]
	Octreotide	SAMA-TOC post liposome modification	111In-DTPA	[112]
	Octreotide	HSPE-PEG4000-OCT	DOX	[113]
	[Tyr3]-Octreotate	Maleimido-TATE	64Cu-DOTA	[114]
	KE108	KE108 post micelle modification via NHS	TDP-A AB3	[115] [116]
	[7–14]BN wild-type	(C18)2-L5-[7–14]BN (C18)2-PEG3000-[7–14]BN	111In-DOTA	[117]
	[7–14]BN-AA1 analogue	MonY-BN-AA1	DOX AUL12 DOX	[118] [119] [120]
	CCK8	(C18)2-L5CCK8	Gd-DOTA/Gd-DTPA	[121]

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
