# Peer review of "Peptide-Based Drug-Delivery Systems in Biotechnological Applications: Recent Advances and Perspectives"

_molecules, 2019, doi:10.3390/molecules24020351_

Round 1

Reviewer 1 Report

The authors summarized recent advances of peptide-based DDS especially using assembled nanostructure the authors have dealt with. Based on the accumulated experiences, the history was precisely summarized. Thus, this reviewer recommends the paper for publication as a review article as it stands.

 Some comment and typographical error were found as follows:

 1)  Page 9-10: As the abbreviation of cell penetrating peptide, “CCP” is sometimes written, should be “CPP”.

 2)  Tumor homing peptides have been developed and applied. This reviewer recommends that the authors discuss this point in the manuscript [e.g. Nat. Commun., 3, 951 (2012); Research on Chemical Intermediates, 44, 4685 (2018)].

Author Response

 1)  Page 9-10: As the abbreviation of cell penetrating peptide, “CCP” is sometimes written, should be “CPP”.

Response:   We have approveded and changed this term.

 2)  Tumor homing peptides have been developed and applied. This reviewer recommends that the authors discuss this point in the manuscript [e.g. Nat. Commun., 3, 951 (2012); Research on Chemical Intermediates, 44, 4685 (2018)].

Response: Our target is not limited to CPP system and we consider both suggest references too specific and not properly to the our review topic. Sorry but we not able to insert our suggestions.

Reviewer 2 Report

The manuscript molecules-419203 covers a very interesting topic, describing the use of peptides in peptide-based drug delivery systems. Mainly, authors focused on the properties and properties that make aggregation of peptides in nanostructures possible. The manuscript is well structured and provides readers with a sequential information.

Overall, the information presented in this review is comprehensive, of interest to the scientific community. However, my main concern is with the overall grammar, writing style. The manuscript needs to be majorly revised or rewritten prior to publication.

Major comments:

The introduction is poor. Contains repeated sentences from abstract and it’s not an appropriate setting for the reviewed information in the manuscript. The conclusion is it too not carefully prepared. The authors should pay more attention to these two sections.

The authors should also pay to other sections for instance section 3.1 and 3.2. Section 3.1 mainly considers TAT peptide, but the title suggests that the authors will present other peptides – CPPs and smart sequences. More information should be added to this section, the authors could even review CPPs  able to cross the Blood-brain barrier. In the literature authors can easily find works of Giralt or Castanho related to the translocation of the BBB. Trans-BBB peptides or BBB shuttles are being studied as peptide-based drug delivery systems into the brain. These CPPs could appear in a table, together with sequence, mechanism and therapeutic application. Again for section 3.2 the authors could present the different peptide-receptor interactions in a table.  It would be easier to follow and compare.

Minor comments/corrections:

Overall, the manuscript is not well written and contains many sentence fragments, choppy sentences, awkward grammar, nonparallel sentence structure, making difficult to read. The reviewer recommends carefully rewriting/revising the manuscript.

1.       INTRODUCTION

The authors report the production of peptides by chemical synthesis but miss the production through DNA recombinant, which is one of the most applied techniques to obtain peptides..

Line

Comment

42

The reference is old. Currently,   there are more recent works reporting the applications of peptides.

43

These advantages depend on the   strategy applied to produce peptides.

44

“easily synthetized”. The authors   are mostly focused on synthetic peptides. It is true that they are and that   this is one of the major advantages of peptides over other biological agents,   such as antibodies. However, they must emphasize that it is a feature of the   synthetic peptides. 

46

Next, they mention the   degradation. It is true. But, it is true for all peptides. This needs to be   clarify.

47

“several chemical approaches” and   in the next sentence they mention some examples. One is the introduction of   specific coded or un-coded amino acids… This is not a chemical modification.   It is obtained by DNA recombinant technology.

54

Needs a reference.

65

“ingredients” ??

2.       Peptide self-assembled nanostructure

2.1   a-helical and b-sheet peptides

This section is confusing

First, it is not formatted properly. Then, an introduction for each secondary structure is missing.

Line

Comment

94

“bring into play”???

109

“take the key role”???

113

“are decisive” – strong statement

2.2   Linear peptides

In this section, authors mention the different nanoparticles that can be obtained by peptides self assemble. It is an exhaustive collection of information with good examples. It is well structured and comprehensible. However, expressions such as, “particular fascinating”, “excellent”, “impressive”, “intelligent”, “contest”, among others are inappropriate…

Line

Comment

119

Can you clarify what is a short   and ultra-short peptide?

120

“particular fascinating”…

131

“impressive”…

133

“they strongly affected” – not   correct

153

“biological marker” – replace   with dye or fluorophore

154

The study was performed in vivo to extrapolate that these   carriers to deliver at constant rates in the body?

155

Very high thermal and metabolic   stability. The sentence shown does not make sense

165

Cut materials

166

“This excellent”…

220

“They”

223

Very recent… Recently…

229

“They”

230

So research is a contest?

233

Very recently… Recently…

256

Very recently… Recently…

256

“unexpected” – the authors are to   bias

2.3   Cyclic peptides

Only a few examples are presented in this section in comparison to the last one. It is well-organized but the mechanisms and some explanations are missing.

Line

Comment

281

“This way” is not correct

284

Which is the mechanism behind the   self-assembling of L and D peptides?

292

Rephrase the sentence. And why   are cyclic peptides not suitable as drug delivery systems?

2.4   Amphiphilic peptides

The section is not well written. It does not have a rational… It lacks explanations, mechanisms, and the presentation of the examples is confused. Some have a lot of information and other only a few words.

Line

Comment

301

Philosophical but does not add   nothing to the text

306

The amino acids are well   presented and a person that work with them quickly understand why they   self-assemble. However, I think that the authors must have a few words   concerning the properties of these amino acids.

308

Physical and chemical properties,   which are…

309

Rephrase

312

In details. I thought that the   authors would explore the self-assemble properties. However, the detail is an   example of the applicability.

324

Rephrase

3.       Self-assembling PAs for targeting in nanostructures

In this section, authors present several examples of peptides capable of forming nanostructures. This nanostructures are then applied in nanotargetting, increase peptide properties,… The examples presented are good and cover a broad line of research. Nevertheless, they are too extensive or only have few comments. In addition, they lack the reason why, mechanisms of action, authors explanations for some evidence. It is a review! In my point of view must not be just a collection of examples. Authors should comment the most interesting evidence, and try to come out with good explanations for some features. The English is difficult to follow…

The authors begin start the section with “In the previous paragraph” if this is a new section it does not make sense to refer to the last paragraph.

Line

Comment

376

“what was preloaded” – terrible   way to talk about the cargoes, drugs, … Rephrase

376

“PAs” comprises an amino acid   sequence “ – well it is a peptide… “which performs” – the sequence does not   perform nothing! The sequence of amino acid residues is responsible for the   targeting and delivery of drugs due to…

378

Not just pathological… and the   diagnosis?

3.1   Cell penetrating peptides (CPPs) and smart sequences

Is not clear why the authors add to the title smart sequences, there was something else to had to this section? Other CPPs either than TAT? CPPs for specific tissues, like brain?

Line

Comment

380

Title has a typo CCPs instead of   CPPs

381

“The first among the tasks”…

382

The reference is from 2002! Since   2002 a lot of CPPs were discovered and studied

383

If they are cationic they have   positive charge – redundant

386

And new ones. One very recent has   the following doi: 10.1016/J.abb.2018.11.010 – I suggest that the authors   read the review and reference it.

387

All the CPPs lack cancer   specificity? In the literature, cannot be found any work reporting CPPs   specificity towards cancer cells?

390

Thermal stimuli, light,   ultra-sound, enzymes, pH. There are more ways of stimuli the release of a   drug. pH and enzymatic are just two examples.

390

Considered as smart is not an   accurate expression in my opinion….

393-402

Very good explanation and   strategy presentation

404

Mices showed

406

Than in controls

3.2   Peptide able to interact with overexpressed receptors

As mention previously the different receptors could appear in a table.

Line

Comment

418

Could allow the intracellular   delivery

419

Of the payload

423

Kept the conformation

427

Conjugation to proteins, lipids,   nanoparticles…

430

New peptide sequences can act –   New and old! That always depend on the purpose of the peptide and on the   applications that you give to the strategy applied

432

1.       They do not act

433

2.       The present

434

The rational design…

441

EPR consists in a phenome where   some molecules are retained in the tumor site due to increase of capillary   permeability. Here you are saying that they increase the blood circulation   and that this feature is due to PEGylation?

445

The spacer allows the maintenance   of molecule flexibility, mobility and increase solubility is some cases

449

Figure 3 (left panel) shows and…

461

Use of enzymes, copper-free   chemistry, …

464

Do you think that these are the   only receptors overexpressed or the only ones that researchers can tackle?

3.2.1         Peptide target for integrins reeptors

Integrins are one of the most important receptors that can be used in active targeting delivery. It is a family composed of many receptors. Thus, in the beginning I think that authors should put a reference to that. Then, they can just focus on the most important ones.

The examples presented are good but it is messy. Example after example after example the reader will be lost… Lack of explanations for some important features is also a problem.

Line

Comment

474-479

Make a better introduction of the   integrin family

477

Literature data… No!

490

In the most cases?

501-508

Confused

516

The authors talk about the three   RGD analogues but then they do not refer specifically to each of them.

517

Compared to other two analogue   sequences. Which analogues?

532-543

Extremely confusing…

3.2.2         GPR Target peptide

GPR are important receptors. They are a family with very different receptors. The authors mention 5 saying that are the most important ones. Why do they talk just about these 5? In the following sections they only specify three. In this brief introduction, I think that the authors should talk a little bit more about the reason why they selected just these three.

3.2.2.1    Somastatin receptors

A number of examples with few comments, information… They are lacking something because I think that the reader does not understand their value.

Line

Comment

550

I did not understood the sentence

552

Rearrange the sentence

555

“Reported in figure 4b” – This is   not correct. Shown in Figure 4b is more accurate

557

“This cyclic peptide is able to   cross the cell membrane via   endocytosis (…)”

567

Every single formulation must be   full characterized. Moreover, every single molecule must be full   characterized.

572

Good explanations

574

The reason why this does not   happen

3.2.2.2    Bombesin receptors

Line

Comment

649

The four receptor-subtytes which   are…

653

The authors do not report nothing   in figures…

660

Just to increase hydrophilicity?   Researchers add PEG to increase other properties rather than solubility

663

Lack fundamentation

672

“Interesting probe” Why?

682

Explain the reasons why authors   performed the substitutions

3.2.2.3    CCK receptors

Line

Comment

698

Rephrase…

4.       Conclusions

The conclusions are not well written… I think that being the paper a review, authors should highlight challenges, enumerate advantages, talk a little bit about what is to come, give a personal comment on what has been done. The last paragraph is poor and “in our guess” is not a proper way to finish such exhaustive paper.
